# Anaesthetic and Perioperative Management of 14 Male New Zealand White Rabbits for Calvarial Bone Surgery

**DOI:** 10.3390/ani9110896

**Published:** 2019-11-01

**Authors:** Mathieu Raillard, Carlotta Detotto, Sandro Grepper, Olgica Beslac, Masako Fujioka-Kobayashi, Benoit Schaller, Nikola Saulacic

**Affiliations:** 1Experimental Surgery Facility (ESF), Department of BioMedical Research, University of Bern, 3008 Bern, Switzerland; carlotta.detotto@dbmr.unibe.ch (C.D.); olgica.beslac@dbmr.unibe.ch (O.B.); 2Vetsuisse Faculty, Department of Clinical Veterinary Medicine, Institute of Anaesthesiology and Pain Therapy, University of Bern, 3012 Bern, Switzerland; 3School of Veterinary Science, Faculty of Science, The University of Sydney, Sydney 2006, Australia; 4Central Animal Facilities, Department for BioMedical Research, University of Bern, 3008 Bern, Switzerland; contact.1352393@campus.unibe.ch; 5Department of Cranio-Maxillofacial Surgery, Inselspital Bern, University Hospital, University of Bern, 3008 Bern, Switzerland; masako.fujioka.kobayashi@gmail.com (M.F.-K.); Benoit.Schaller@insel.ch (B.S.); nikola.saulacic@insel.ch (N.S.)

**Keywords:** rabbits, New Zealand White, calvarial bone, craniotomy, anaesthesia, analgesia, V-gel^®^, pain, pain score, rabbit grimace

## Abstract

**Simple Summary:**

Bone substitutes are commonly used when bone grafts are necessary in human craniofacial surgery. To study the properties and biologic behaviour of those substitutes, they can be implanted in experimental animals. A frequently used model involves the creation of critical-sized defects (defects that are too large to heal by themselves) in the rabbits calvaria (the top part of the skull). The procedure was initially described in 1989 and the authors considered and reported that post-operative analgesia was not necessary. In our experience, this procedure is invasive and can result in severe postoperative pain. The anaesthetic management of rabbits undergoing this procedure is challenging. Most of the recent publications using this model fail to provide sufficient information on perioperative animal management. With this report we have aimed to document a possible practical and simple anaesthetic and postoperative management of rabbits undergoing this procedure. Particular emphasis has been placed on postoperative pain assessment, duration and treatment.

**Abstract:**

Calvarial bone surgery on rabbits is frequently performed. This report aims to document a simple and practical anaesthetic and perioperative management for this procedure. Fourteen male New Zealand white rabbits were included in the study. Subcutaneous (SC) dexmedetomidine, ketamine and buprenorphine ± isoflurane vaporized in oxygen administered through a supraglottic airway device (V-gel^®^) provided clinically suitable anaesthesia. Supplemental oxygen was administered throughout recovery. Monitoring was clinical and instrumental (pulse-oximetry, capnography, invasive blood pressure, temperature, arterial blood gas analysis). Lidocaine was infiltrated at the surgical site and meloxicam was injected subcutaneously as perioperative analgesia. After surgery, pain was assessed five times daily (composite behavioural pain scale and grimace scale). Postoperative analgesia included SC meloxicam once daily for four days and buprenorphine every 8 h for three days (unless both pain scores were at the lowest possible levels). Rescue analgesia (buprenorphine) was administered in case of the score > 3/8 in the composite pain scale, >4/10 on the grimace scale or if determined necessary by the caregivers. Airway management with a V-gel^®^ was possible but resulted in respiratory obstruction during the surgery in two cases. Hypoventilation was observed in all rabbits. All rabbits experienced pain after the procedure. Monitoring, pain assessments and administration of postoperative analgesia were recommended for 48 h.

## 1. Introduction

Rabbits are a standardized animal model applied in the research of biomaterials used in bone regenerative medicine [1,2,3]. One commonly performed surgical procedure is the creation of defects in the calvarial bone [4,5,6,7,8]. An early description of this surgery stated that “postoperative analgesia was not considered necessary” [9]. Recent publications using this model fail to provide sufficient information on perioperative animal management [2,3,4,5,6,7,8,9,10,11,12,13,14,15,16,17,18,19,20,21,22,23,24] and do not comply with ARRIVE (Animal Research: Reporting *In Vivo* Experiments) guidelines [25]. It is possible that many experimental rabbits undergoing this procedure receive inappropriate perioperative analgesia. In our experience, this procedure is invasive and can result in severe postoperative pain. Ethical concerns on the lack of analgesia in experimental rabbits undergoing surgery have already been raised [26] and analgesia recommendations for this surgical model were not found in the literature.

This manuscript aims to report a simple and practical anaesthetic and perioperative management of New Zealand White rabbits undergoing calvarial bone surgery. Emphasis was placed on the evaluation of postoperative pain and the description of an acceptable analgesia protocol.

## 2. Materials and Methods

### 2.1. Animals and Husbandry

All procedures were carried out with the approval of the Animal Care Committee of the Canton of Bern, Switzerland (permission BE89/17) in accordance with current Swiss legislation on Animal experimentation.

Fourteen 17-week old male New Zealand White Rabbits (Charles Rivers Laboratories, Romans, France) entered this study. They were housed individually in suspended cages (Cage body P type, 970 × 895 × 1718 mm) and acclimatised for three weeks prior to the start of the experiment. To avoid stereotypies, the space between two cages was left open permitting animal contact. In case of conflict, a transparent separator was inserted between two cages to maintain visual contact. Each cage was provided with an elevated platform, a wood block and a chew ball as enrichments. Animals were kept on a 12:12 light:dark cycle (lights on at 06:30), room temperature was kept between 19–21 °C and humidity at 45 ± 10 %, in compliance with GV-Solas guideline. Each rabbit received 200 g of standard rabbit food (Provimi Kilba, Aargau, Switzerland) per day and water ad libitum.

### 2.2. Anaesthetic Management

The rabbits were moved from the housing unit (Central Animal Facilities) to the Experimental Surgery Unit in individual cages the morning of the surgery (both facilities on the same campus). Physical examination was performed by a trained veterinarian and animals were weighed on electric scales. Dexmedetomidine 100 mcg kg^−1^ (Dexdomitor^®^; Provet AG, Lyssach, Switzerland), ketamine 15 mg kg^−1^ (Narketan^®^, Vetoquinol AG, Bern, Switzerland) and buprenorphine 30 mcg kg^−1^ (Temgesic^®^, Rechitt Benckiser, Wallosellen, Switzerland) were mixed in a 2 mL syringe and injected subcutaneously (SC) in the neck area with a 23 G needle. After the injection, the rabbits were left undisturbed for 10 to 12 min. Sedation was assessed and recorded by a single operator using a simple descriptive scale that has been reported previously [27] (Table 1). Animals were taken out of their cages and positioned in sternal recumbency on the preparation table. A small amount of eye unguent (Bepanthen Augen und Nasenalbe, Bayer Vital GmbH, Leverkusen, Germany) was applied over the corneas, the eyelids were gently massaged to ensure lubrication of the eyes, heads positioned in a non-tight facemask and pure oxygen was administered at 2 L min^−1^. The ears were clipped and disinfected and a 22 G intravenous catheter (Vasofix^®^ Safety, Bbraun, Melsungen AG, Germany) was inserted into the left marginal vein. A 22 G catheter (Vasofix^®^ Safety, BBraun Melsungen AG, Germany) was inserted into one of the auricular arteries for invasive blood pressure monitoring and to facilitate blood sampling during the procedure. The heads were clipped and aseptically prepared for surgery.

The rabbits were moved to the operating theatre and positioned in sternal recumbency in a plastic trough with open extremities ensuring a stable position without pressure on the chest and an accessible head. After further pre-oxygenation via a facemask, a V-gel ^®^ (Docsinnovent Ltd., London, UK) connected to a side stream capnography monitor (Datex-Ohmeda S3, GE Healthcare Inc., Helsinki, Finland) was inserted by a single trained operator. The correct positioning of the device was assessed through visualization of successive CO_2_ curves on spontaneous breaths.

The ease of insertion of the supraglottic device was scored using a modification of a previously published simple descriptive scale [28] (Table 1). In case of score of 1 or 2, further ketamine was administered intravenously (IV) to effect by 0.5 mg kg^−1^ increments. If administered, the ketamine dose was recorded. The rabbits were connected to a Mapleson F system (Jackson Rees modified T-piece with the distal end of the bag connected to a fluoabsorber) and 1.5 to 2.5 L minute^−1^ of oxygen was administered throughout the procedure. Isoflurane (0.5 to 2%) (Forene, Abbvie AG, Baar, Switzerland) vaporized in oxygen was added if necessary and adjusted according to anaesthetic depth.

Perioperative analgesia included the buprenorphine administered with the premedication, meloxicam (0.3 mg kg^−1^) (Metacam^®^, Boehringer Ingelheim, Basel, Switzerland) injected SC in the neck area and lidocaine 1% (1 mL) (Bichsel, Interlaken, Switzerland) infiltrated subcutaneously at the surgical site before placement of the surgical drapes (approximately 3 to 5 min before surgery started).

Intravenous fluid administration (NaCl 0.9%, 5 mL kg^−1^ h^−1^; B. Braun Medical AG, Sempach, Switzerland) was initiated at the time of IV catheterization and continued throughout the procedure. Perioperative antimicrobial prophylaxis consisted of procaine penicillin 150,000 IU mL^−1^ + benzathine penicillin 150,000 IU mL^−1^ (Duplocillin ^®^, MSD Animal Health, Luzern, Switzerland), 0.01 mL kg^−1^ SC before surgery.

Monitoring of anaesthesia was: (1) clinically conducted by a veterinary anaesthetist and (2) instrumentally conducted using a multiparametric anaesthesia monitor (Datex-Ohmeda S3, GE Healthcare Inc., Helsinki, Finland), which included: (i) pulse-oximetry (SpO_2_), with a probe placed on one of the digits of the pelvic limb, (ii) capnography (P_E’_CO_2_) and gas analysis connected to the appropriate port of the V-gel^®^, (iii) invasive blood pressure (correctly configured and zeroed transducer placed at the height of the heart and connected to the arterial catheter in an auricular artery) and (iv) rectal temperature. Physiologic variables were recorded at 5 min intervals throughout the procedure on a standard anaesthesia record made available online by the Association of Veterinary Anaesthetists (AVA) [29]. Isoflurane requirement (vaporizer setting) was also recorded. Approximately half way through the procedure, arterial blood gas analysis was performed. In the last four rabbits, additional arterial blood gases were analysed immediately after the arterial catheter placement before the beginning of the procedure approximately 15–20 min after the injection of the premedication drugs.

### 2.3. Surgical Procedure

Skin was incised on a 3 cm straight line from the nasal bone to the midsagittal crest. Periosteum was elevated to allow exposure of the parietal bone. Two bone defects (10 mm) on each side of the midline were created with a trephine bur under irrigation with sterile saline. Care was taken to avoid injury of the dura. A drilling procedure with a trephine bur was done very carefully with low speed (maximum 800 rpm) by trained cranio-maxillofacial surgeons. Drilling was stopped before reaching the dura and the bone pieces were removed by blunt instruments such as mucosa elevators. Under the conditions of the experiment, fresh blood needed to be mixed with the biomaterial under investigation. Blood was therefore sampled from the catheterized auricular artery approximately half way through the procedure (arterial blood gases were analysed at the same time). Following mixture, the biomaterials were implanted directly into the bone defects. After implantation the periosteum and skin were closed with interrupted sutures in layers using 4–0 Vicryl and 4-0 Monocryl sutures (Ethicon, Somerville, NJ, USA). The wound surface was further protected with spray film dressing (Opsite^®^ spray, Smith & Nephew, London, UK).

### 2.4. Postoperative Management

At the end of the procedure isoflurane administration was discontinued in case it had been initiated. The arterial cannula was removed and gentle compression was applied for 3 to 5 min. The V-gel^®^ was removed. The duration of anaesthesia, surgery and any eventual perioperative complications were recorded.

Clinical monitoring was continued during the recovery period. The eyes were regularly lubricated. Oxygen was administered by flow at 2 L minute^−1^ after removal of the supraglottic device. The rabbits were placed on a soft mattress under infrared lights (150 Watts at approximately 50 cm) to provide an external source of heat. Rectal temperature and SpO_2_ were regularly measured and skin temperature was subjectively assessed. Intravenous fluids administration was continued at 3 to 5 mL kg^−1^ h^−1^ until removal of the vascular access (IV catheters were removed after return of righting reflex). The time to return of righting reflex was recorded. Once fully recovered (bright, alert, responsive, up to temperature, spontaneously moving around in their transport cage), the rabbits were returned to their facility.

Pain was assessed using a composite behavioural pain scale and the evaluation of the rabbit grimace scale (Table 2) [30,31]. Baseline scores were obtained using this evaluation during the three days prior to surgery. Pain was assessed at standard time points before any drug administration by one of three trained operators (06:00 am, 10:00 am and 14:00, S.G. 18:00, C.D.; 22:00, M.R.) on the day of surgery and for the following three days (day 1, day 2 and day 3). The operators knew the rabbits involved in this experiment, were familiar with the setting and the scores used and had experience in pain assessment and recognition of rabbits and other species. Postoperative analgesia included meloxicam 0.3 mg kg^−1^ SC once daily (10:00 am) for 4 days in all cases and buprenorphine 20–30 mcg kg^−1^ (0.3 mL of Temgesic 0.3 mg mL^−1^/rabbit) SC three times daily (06:00 am, 14:00 and 22:00) for three days after surgery unless both pain scores (composite and grimace) were at the lowest possible levels (0/8 and 0/10). Rescue analgesia (SC buprenorphine 20–30 mcg kg^−1^: 0.3 mL of Temgesic 0.3 mg mL^−1^ / rabbit) was administered at 10:00 am and 18:00 in case of a score of more than 3 in the composite pain scale, more than 4 on the grimace scale or if subjectively considered necessary by the caregivers, mostly based on behavioural assessment, despite scores lower than the cut-off values. Pain scores and drugs administered were recorded on a standard hospitalization form.

Particular attention was given to the overall behaviour of the rabbits and water and food intakes, presence of urine and faeces in the litter (faecal output was monitored). Abdomens were palpated each day. Animals were weighed 48 h after surgery. Weight loss of >10% was considered as an end-point.

### 2.5. Euthanasia

Between one and six months after the surgery, the rabbits were euthanised. After sedation with SC ketamine 65 mg kg^−1^ (Narketan^®^, Vetoquinol AG, Bern, Switzerland) and xylazine 4 mg kg^−1^ (Xylazin 20 mg mL^−1^, Streuli Pharma AG, Uznach, Switzerland), pentobarbital sodium 150 mg kg^−1^ (Esconarkon^®^, Streuli Pharma AG, Uznach, Switzerland) was administered intravenously.

GraphPad Prism 7 for Mac OS X (GraphPad Software, La Jolla, CA, USA) was used for the graphical representation of simple descriptive statistics. No statistical test was applied to this data set.

## 3. Results

The mean ± SD weight before surgery was 3.3 ± 0.4 kg. The median (range) sedation score was 13 (10–13). Individual sedation scores are presented in Table 3. Insertion of the V-gel was easy and quick in most cases: the median (range) quality of the induction score was 4 (2–4) although two rabbits required additional ketamine (0.5 mg kg^−1^ IV). Those rabbits were the two rabbits who scored lower sedation scores (Rabbits 1 and 11) and were the first rabbits to be anaesthetized on the first and third day of procedures. A R4 V-gel^®^ was inserted into all but one rabbit (Number 11). That rabbit had a smaller head compared to the others and the R4 V-gel^®^ was too large to be advanced smoothly so a R3 V-Gel^®^ was placed. Despite pre-oxygenation, desaturation happened in several cases during airway management. In two cases, obvious respiratory obstruction developed during the early stages of surgery (increased inspiratory and expiratory efforts and inconsistent P_E’_CO_2_ values). The V-gels were promptly removed and oxygen +/− isoflurane were administered via a tight facemask connected to the Mapleson F breathing system for the rest of the procedure.

Individual isoflurane requirements are reported in Table 3. Isoflurane was not administered to 5/14 rabbits and the concentration administered was less than 1% in six others. The mean arterial pressure (MAP) was generally between 60 and 80 mmHg although transient readings between 55 and 60 mmHg were observed in some rabbits. Individual MAP, pulse rate and P_E’_CO_2_ ranges are reported in Table 3. The blood gas could not be analysed in one animal as the arterial catheter was accidentally dislodged before blood could be withdrawn; gentle compression was applied. Table 4 presents individual arterial blood gas analysis. Marked hypercapnia (PaCO_2_ of 59 to 79 mmHg) developed in all but the first rabbit. Arterial oxygen tension was between 99 and 407 mmHg in all but two animals: rabbit 3 before insertion of the V-Gel^®^ (68 mmHg) and rabbit 7 which had a facemask at the time of blood sampling (77 mmHg). Base excess was between 0 and 21 and blood glucose concentration was elevated (between 10 and 21 mmol L^−1^). The individual duration of anaesthesia (55 to 80 min), surgery (30 to 45 min) and return of righting reflex (20 to 55 min) are reported in Table 3.

The evolution of the composite pain score and grimace scale in the three days following surgery are presented in Figure 1A,B. All rabbits exhibited signs of pain in the post-operative period. Both pain scores were back to their baseline values approximately 48 h after surgery. Figure 2 presents the number of animals receiving buprenorphine at each evaluation time point, either scheduled (06:00 am, 14:00 and 22:00) or rescue (10:00 am and 18:00). All rabbits received buprenorphine every 8 h on the day of surgery and 11 of them required rescue analgesia. Buprenorphine administration decreased from day one after surgery. Weight loss was observed in all rabbits 48 h after surgery (3 to 6%).

The dura in the right bone defect of rabbit 14 was partially damaged during surgery resulting in moderate bleeding. Recovery from anaesthesia was slower than in the other animals. Mentation was reduced in the first hours after surgery and the animal developed head tilting and walking on the circle. After 48 h of intensive observation and appropriate analgesia administration, this condition did not improve. Weight loss was around 15%, therefore the rabbit was euthanized.

## 4. Discussion

The description of the presented anaesthetic and perioperative management raises several clinically relevant aspects. First, the use of the V-gel^®^ in this surgical procedure had the potential to lead to respiratory obstruction. Second, hypoventilation and ventilation/perfusion inequalities were documented in all animals and may have been unobserved without arterial blood gas analysis. Third, despite early multimodal analgesia, rabbits undergoing this surgical procedure did show signs of pain in at least the first 48 h after surgery. This is particularly 3R relevant as post-operative analgesia of rabbits undergoing this surgical model is inconsistently reported.

The V-gel^®^ placement was difficult in the two less sedated animals. Although V-Gel^®^ insertion was easy and successful in the other rabbits, respiratory obstruction developed in the early surgical stages of two rabbits. Endotracheal intubation can be challenging in rabbits [32] although airway management is recommended for long well-controlled periods of anaesthesia [33]. The V-Gel^®^ is specifically designed for rabbits [34] and has been shown to be a practical alternative to endotracheal intubation and a laryngeal mask for airway management and controlled mechanical ventilation [35] potentially causing less tissue damage than endotracheal intubation [36]. The V-Gel^®^, however, has the potential to cause laryngeal compression without an obvious clinical sign [35]. The respiratory obstruction observed in two out of 14 rabbits could be due to an incorrect placement of the device or dislodgement during craniotomy; this highlights the importance of perioperative monitoring.

Arterial blood gas analysis documented ventilation–perfusion inequalities and lower arterial oxygen tension than what could be expected in a small animal that is administered pure oxygen. The rabbits enrolled in this experiment were obtained from a colony free of respiratory pathogens and appeared clinically healthy. Hypoxaemia with a similar anaesthesia protocol without buprenorphine has been documented in rabbits [37]. The low arterial oxygen tension could therefore be an effect of our anaesthesia protocol. However, it may also suggest that the V-gel^®^ was not completely sealing the airway. This finding would have been unobserved had blood gases not been analysed as SpO_2_ was > 95% throughout the procedure in all rabbits. This underlines the fact that oxygen administration in the perianaesthetic context is reasonable in rabbits.

A sidestream capnograph and relatively high fresh gas flows were used so differences between P_E’_CO_2_ and arterial PaCO_2_ are not surprising. Superiority of arterial blood gas analysis over P_E’_CO_2_ monitoring was already previously established in pet rabbits [38]. In all cases, hypoventilation was documented approximately 45 min after sedation. Our anaesthesia protocol was adapted from current literature [28,37,39,40,41,42]. A previous study investigated the effects of 0.03 mg kg^−1^ of buprenorphine in rabbits one hour before the administration of a similar anaesthesia protocol (SC medetomidine 0.25 mg kg^−1^ and ketamine 15 mg kg^−1^) [40]. It was found that recoveries were longer but no respiratory depression was observed 10 min after the administration of medetomidine and ketamine. In our experiment, less alpha-2 agonist was used but buprenorphine was administered at the same time as the other drugs. PaCO_2_ measured in the last four rabbits at the beginning of anaesthesia (approximately 15–20 min after injection) was lower than PaCO_2_ during the surgery but higher than what was reported by Murphy et al. [40]. It is possible that simultaneous administration of all the drugs increased respiratory depression. On the other hand, hypoventilation might have been unobserved in some of the previous studies as blood gas analysis was performed fairly early after the administration of the drugs. Its clinical relevance is still unknown.

Anaesthetic risk is higher in rabbits (overall 1.39%) than in other domestic species [43]. Perianaesthetic mortality mostly occurs after recovery from anaesthesia (64%) [43]. Many reasons can explain this, but it is likely that the lack of monitoring plays a major role. Hypoxaemia is frequent with injectable protocols; our previous experience documented frequent desaturation in the recovery phase. Because of their high surface/volume ratio, rabbits are prone to developing hypothermia during anaesthesia [44]. Continuous monitoring, pain assessment, warming and oxygen supplementation seem essential.

Despite early multimodal analgesia (opioid, non-steroidal anti-inflammatory drug and topical local anaesthetic agent) every animal enrolled in this experiment experienced pain. This was a major finding and contrasts the reports on the early application of this technique [25]. This may originate from an improved pain assessment [30,45,46,47]. Scientists should be aware that the calvarial bone defect model in rabbits may result in strong acute pain and that monitoring and analgesia administration after this surgery is essential. Using a composite pain scale and the rabbit grimace scale allowed us to identify times during the postoperative phase in which rabbits were likely to be in pain and require additional analgesia. Pain reached a peak during the 12 h following surgery and the following day. Analgesia was necessary every 4 to 8 h on the day of surgery and after 1.5 days; thereafter, most rabbits appeared comfortable.

This study has several limitations. First, it is well established that the pain response is sex-dependent [48]. The findings of the present report might thus apply to male rabbits only. Second, animals were not filmed between pain assessments. Being prey species, it is possible that the presence of an observer modified their behaviour and altered pain scores and expression. Third, the descriptive pain scale used was not validated. It is however simple and rapid to use and allowed for the establishment of cut off scores similar between operators. Fourth, a descriptive study is by nature not blinded. Fifth, because of the low number of included animals, it is possible that some of the complications were over-represented and that some other possible complications were not encountered.

The clinical relevance of the documented hypercapnia is unknown. Although outcomes were mostly good from an anaesthesia point of view, peri-anaesthetic management can still be improved. Future studies will look at investigating the impact of opioids and other factors on hypoventilation during anaesthesia and will aim to further refine the post-operative analgesia protocol.

## 5. Conclusions

Within its limitations, the present report describes a simple and practical anaesthetic management for rabbits undergoing calvarial bone surgery. Subcutaneous administration of dexmedetomidine, ketamine and buprenorphine ± addition of low-dose isoflurane provided clinically suitable anaesthesia of sufficient duration. Nevertheless, it resulted in marked hypoventilation of unknown clinical importance. Airway management with a V-gel^®^ was possible, but the continuous intraoperative monitoring is important due to a potential respiratory obstruction. Rabbits will experience pain after this surgical procedure and we recommend maximum postoperative monitoring and regular administration of postoperative analgesia for 48 h after surgery. This is particularly 3R relevant as post-operative analgesia of rabbits undergoing this surgical model is inconsistently reported.

## Figures and Tables

**Figure 1 animals-09-00896-f001:**
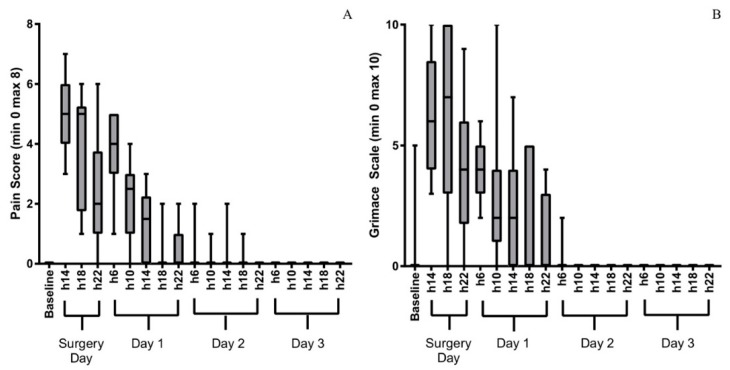
Evolution of composite pain scores (**A**) and the grimace scale (**B**) of 14 rabbits undergoing surgical creation of critical-sized defects in the calvarial bone assessed by one of three trained operators at baseline, on the day of surgery and for 3 days after intervention at 06:00 am, 10:00 am, 14:00, 18:00 and 22:00. Minimum composite pain score is 0, maximum is 8; lowest grimace score is 0, maximum is 10. The boxes represent the values from the lower to the upper quartile, the ends of the whiskers the minimum and maximum values. The median values are represented by the lines in the boxes.

**Figure 2 animals-09-00896-f002:**
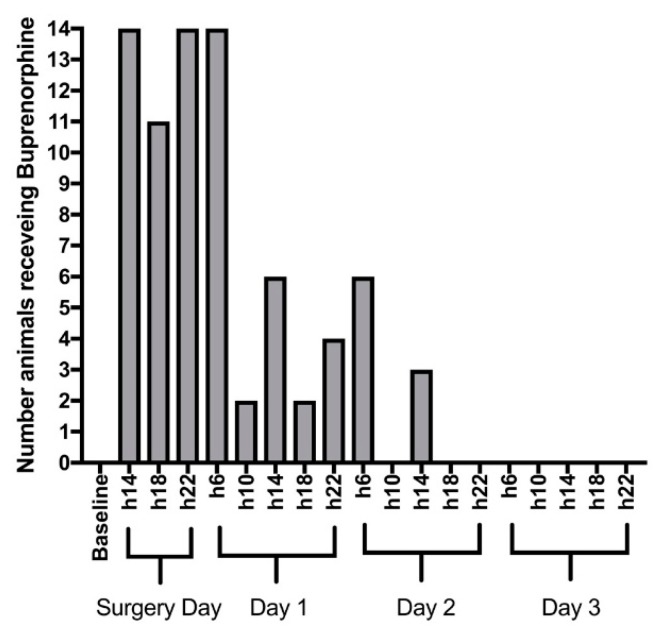
Number of rabbits (out of 14) administered buprenorphine after each assessment (06:00 am, 10:00 am, 14:00, 18:00 and 22:00), after surgical creation of critical-sized defects in the calvarial bone on the day of surgery and for 3 days after intervention at 06:00 am, 10:00 am, 14:00, 18:00 and 22:00. The top of the grey bars represents the number of animals treated. Postoperative analgesia included meloxicam once daily (10:00 am) for 5 days in all cases and buprenorphine three times daily (6:00 am, 14:00 and 22:00) for three days after surgery unless both pain scores (composite and grimace) were at the lowest possible levels. Buprenorphine (rescue analgesia) was administered at 10:00 am and 18:00 if necessary.

**Table 1 animals-09-00896-t001:** Sedation scale [27] and quality of induction and condition for V-Gel^®^ insertion modified from Grint and Murison [28] used after anaesthesia agents administration in 14 rabbits undergoing surgical creation of critical-sized defects in the calvarial bone.

Sedation Scale in Rabbits [27]		
Variable	Behaviour of the rabbit	Score
Spontaneous posture	Normal	0
	Sedated but standing/sitting with head up	1
	Lying sternally head up	2
	Lying sternally head down	3
	Lying laterally, responding to stimuli	4
	Lying laterally, not moving when stimulated	5
Resistance to being placed dorsally recumbent	Strong/normal resistance	0
	Moderate resistance	1
	Slight resistance	2
	No resistance	3
Jaw relaxation	Normal tonus	0
	No resistance to mouth opening	1
Palpebral reflex	Normal	0
	Decreased	1
	Absent	2
Limb withdrawal in response to pinching	Normal	0
	Decreased	1
	No reaction	2
Total sedation score		
**Quality of Induction and Condition for V-gel^®^ Insertion Modified from Grint and Murison [28]**		
Description		Score
Excitement, and some struggling, requires additional intravenous ketamine to permit V-gel^®^ insertion		1
Some excitement, vigorous swallowing, additional ketamine needed to permit V-gel^®^ insertion		2
Smooth induction but some swallowing and resistance to V-gel^®^ insertion		3
Very smooth induction of anaesthesia, no swallowing during V-gel^®^ insertion		4

**Table 2 animals-09-00896-t002:** Composite behavioural pain scale and rabbit grimace scale [30,31] used by one of three trained operators to assess pain in 14 rabbits undergoing surgical creation of critical-sized defects in the calvarial bone at baseline, on the day of surgery and for 3 days after intervention at 06:00 am, 10:00 am, 14:00, 18:00 and 22:00.

Composite Behavioural Pain Scale		
Variable	Behaviour of the Rabbit	Score
Attitude	Active, interested	0
	Calm, move when touched	1
	Immobile, consolidated, uninterested	2
Appetite	Normal, eating and drinking	0
	Only drinking or only eating	1
	Not eating or drinking	2
Mobility	Mobility is normal	0
	Pain during action	1
	Inactive	2
Appearance	Straight, bright pelt	0
	Dull, slightly ruffled fur	1
	Ruffled fur	2
Total score	(Best = 0, Worst = 8)	
**Rabbit Grimace Scale [29,30]**		
Orbital Tightening		0
		1
		2
Cheek Flattening		0
		1
		2
Nose Shape		0
		1
		2
Whisker Position		0
		1
		2
Ear Position		0
		1
		2
Total Pain score	(Best = 0, Worst = 10)	

**Table 3 animals-09-00896-t003:** Individual preoperative weight, sedation scores, quality score of induction and V-gel^®^ insertion, pulse rate, mean arterial pressure, end tidal carbon dioxide pressure, anaesthesia and surgery durations and return of righting reflex after anaesthesia in 14 rabbits undergoing surgical creation of critical-sized defects in the calvarial bone.

Rabbit	1	2	3	4	5	6	7 *	8	9	10	11 *	12	13	14
Weight	3.7	3.2	3.8	3.3	3.5	3.4	3.4	3.3	3.5	3.6	3.5	3.3	3.5	3.3
Sedation score	10	13	13	13	13	12	13	13	13	13	10	13	13	13
Quality score	2	4	4	4	4	4	3	4	4	4	2	4	4	4
V-gel^®^	R4	R4	R4	R4	R4	R4	FM	R4	R4	R4	R3	R4	R4	R4
% iso	1–2.5	0.5-1.5	0–0.5	0.2–1	0.5–1.0	0.2–1.2	0	0	0.2–1	0–0.5	0–1	0	0	0
PR (minx–max) beats minute^−1^	200–250	204–221	195–214	151–182	128–157	155–216	174–198	130–184	134–175	170–183	154–208	169–187	132–165	170–185
MAP (min–max)(mmHg)	55–65	72–77	57–74	64–74	69–77	61–79	64–77	59–76	64–82	58–67	56–82	69–83	67–75	61–68
P_E’_CO_2_ (min–max)(kPa)	5.3–7.5	4.7–7.2	5.2–8.3	5.3–6.1	4.3–6.5	5.5–8.3	8.4–10.8	6.7–9.7	6.1–8.7	5.0–7.0	3.3–NA	5.6–6.8	4.8–6.4	5.0–6.7
P_E’_CO_2_ (min–max)(mmHg)	40–56	35–54	39–60	40–46	32–49	41–62	63–81	50–73	46–65	38–53	25–NA	42–51	36–48	38–50
Anaesth. Duration (min)	80	65	75	60	65	60	65	70	70	70	85	65	55	60
Surgery duration (min)	45	30	30	35	35	30	35	40	40	30	40	30	30	30
Return righting reflex (min)	20	45	35	35	35	25	25	45	25	25	20	40	35	55

FM: Facemask; MAP: Mean arterial pressure; PR: Pulse rate; P_E’_CO_2_: End tidal carbon dioxide. * Rabbits 7 and 11 had a facemask during surgery because of respiratory obstruction with V-gel^®^.

**Table 4 animals-09-00896-t004:** Intraoperative individual arterial blood gas analysis of 13 rabbits (1-3-4-5-6-7-8-9-10-11/2-12/2-13/2-14/2) undergoing surgical creation of critical-sized defects in the calvarial bone, approximately 45 min after administration of medetomidine, ketamine and buprenorphine subcutaneously with or without isoflurane. Additional arterial blood gas was analyzed in rabbits 11, 12, 13 and 14 approximately 15 to 20 min after injection (11/1-12/1-13/1-14/1).

Rabbit	1	2	3	4	5	6	7 *	8	9	10	11/1	11/2 *	12/1	12/2	13/1	13/2	14/1	14/2
pH	7.29	NA	7.36	7.39	7.40	7.41	7.30	7.27	7.35	7.35	7.46	7.35	7.23	7.24	7.34	7.29	7.31	7.36
P_a_O_2_ (kPa)	13.2	NA	31.1	37.3	49.7	48.3	10.3	39.1	43.4	47.3	54.3	45.3	25.3	43.1	9.1	50	23.3	43.6
P_a_O_2_ (mmHg)	99	NA	233	280	373	362	77	293	306	355	407	340	190	323	68	375	175	327
P_a_CO_2_ (kPa)	4.8	NA	9.3	8.8	9.9	8.8	8.8	10.5	10.0	8.3	6.0	9.1	8.9	10.5	7.9	9.9	7.9	9.2
P_a_CO_2_ (mmHg)	36	NA	70	66	74	66	66	79	75	62	45	68	67	79	59	74	59	69
BE	16	NA	13	14	21	18	7	10	16	8	9	12	0	6	6	9	4	14
Glucose (mmol L^−1^)	20	NA	21	18	17	18	15	15	16	17	13	24	10	15	13	20	10	19
Lactates (mmol L^−1^)	1.4	NA	1.0	1.0	0.9	1.1	0.5	1.1	1.3	0.7	1.9	1.2	0.9	0.5	1.9	1.0	2.5	0.6

BE: Base excess; NA: Not available P_a_O_2_: Arterial partial pressure of oxygen; P_a_O_2_: Arterial partial pressure of carbon dioxide. * Rabbits 7 and 11 had a facemask during surgery because of respiratory obstruction with V-gel^®^.

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
