# Peer review of "Anaesthetic and Perioperative Management of 14 Male New Zealand White Rabbits for Calvarial Bone Surgery"

_animals, 2019, doi:10.3390/ani9110896_

Round 1

Reviewer 1 Report

Well written manuscript. Better to develop the Introduction in a logical way to give a better idea of the importance of the study.

creation of critical-sized defects in the rabbits calvaria is invasive and result in severe postoperative pain.

This research question is very important in pain management of laboratory animals. The interesting point is the post-operative pain.

Comments to improve the methodology

Line 86- Explain how did the eyes were lubricate (the procedure)

Line 90—91- Give the purpose of catheter inclusion in to the auricular arteries.

Line 129- 131- explain why did you analyze additional atrial blood gases in the last 4 rabbits.

Line 139- Explain what measures did you take to avoid injuries to the dura

Line 149- why did you place the rabbit under infra-red light, explain the purpose here.

Line 151-152- check grammar

Line 152-153- what were the criteria to detect fully recover of rabbit, explain here

Line 156-157- Do these time periods after the surgery was completed? After fully recovered?

Line 215- What is the definition of immediate post operative period here? Is your first pain assessment time, better to give the time in ours/minutes

Author Response

Dear Reviewer,

Thank you for your time and your review. We do appreciate very constructive comments. Please find our revised manuscript as well as a detailed answer to your comments. We hope our changes will match your expectations.

Best regards,

Line 86- Explain how did the eyes were lubricate (the procedure)

The sentence was modified as follows: “A small amount of eye unguent (Bepanthen Augen und Nasenalbe, Bayer Vital GmbH, Leverkusen, Germany) was applied over the corneas and eyelids were gently massaged to ensure eyes lubrication”

Line 90—91- Give the purpose of catheter inclusion in to the auricular arteries.

The sentence was modified as follows: “A 22 G catheter (Vasofix® Safety, BBraun Melsungen AG, Germany) was inserted in one of the auricular arteries for invasive blood pressure monitoring and to facilitate blood sampling during the procedure.”

Line 129- 131- explain why did you analyze additional atrial blood gases in the last 4 rabbits.

Thank you for this comment. This made us realise that the table 4 with arterial blood gas analysis was actually missing. It has been added to the text.

Hypoventilation was greater and PaO2 was lower than what we expected in all first 10 rabbits. This would have been unnoticed without the arterial blood gas analysis. We wanted to assess whether this developed with time or early after the sedation and if it was of similar extent at the peak of the sedative effect. Also, a recent abstract from Montreal suggested that high FiO2 in rabbits could lead to hypoventilation so we wanted to have a baseline without oxygen. We find the results relevant but were concerned that adding the rational would add confusion to the reader. Would the reviewer prefer to see this information added to the manuscript?

Line 139- Explain what measures did you take to avoid injuries to the dura

The following sentence was added: “Drilling procedure with a trephine bur was done very carefully with low speed (maximum 800 rpm) by trained cranio-maxillofacial surgeons. Drilling was stopped before reaching the dura and the bone pieces were removed by blunt instruments such as mucosa elevators.”

Line 149- why did you place the rabbit under infra-red light, explain the purpose here.

The sentence was modified as follows: “Rabbits were placed on a soft mattress under infrared lights (150 Watts at approximately 50 cm) to provide an external source of heat”

Line 151-152- check grammar

The sentence was modified as follows: “Intravenous fluids administration was continued at 3 to 5 mL kg-1 h-1 until removal of the vascular access (IV catheters were removed after return of righting reflex).”

Line 152-153- what were the criteria to detect fully recover of rabbit, explain here

The sentence was modified as follows: “Once fully recovered (bright, alert, responsive, up to temperature, spontaneously moving around in their transport cage), rabbits were returned to their facility.”

Line 156-157- Do these time periods after the surgery was completed? After fully recovered?

The assessment happened at fix hours where the rabbits were housed; rabbits were assessed if they had already returned from surgery after recovery, otherwise they were still being monitored by the anaesthesia team. Would the reviewer like this information to be added to the text?

Line 215- What is the definition of immediate post operative period here? Is your first pain assessment time, better to give the time in ours/minutes

The reviewer is right and the word “immediate” was removed from the manuscript as it was imprecise and ambiguous. The time in minutes/hours cannot be given with precision as surgeries finished at different times but assessments were fix; however, signs of pain were observed during the first 48 hours.

Reviewer 2 Report

Very Well written paper! Well done!

Describes in great detail the procedure.

They have taken great care to look for pain and provide rescue analgesia when required.

To me line 136 (new sentence, Bloods were sampled ...) is no clear. Perhaps the authors should clarify or rewrite the sentence?

The tables depending on the journal may need a bit of formatting, especially table 3.

From their results it is quite apparent that the V-gels were easier to place then intubating rabbits.

From our experience sometimes they do not produce a tight seal around the larynx (leak) and may make ventilation difficult.

Perhaps one of the suggestions for future procedures would be intubating rabbits for these kind of procedures. This is what would be done in clinics.

Author Response

Dear Reviewer,

Thank you for your time and your very positive review. Please find our revised manuscript as well as a detailed answer to your comments. We hope our changes will match your expectations.

Best regards,

To me line 136 (new sentence, Bloods were sampled ...) is no clear. Perhaps the authors should clarify or rewrite the sentence?

Thank you for this comment. The reviewer is right. Sentence has been modified as follows: “Under the conditions of the experiment, fresh blood needed to be mixed with the biomaterial under investigation. Blood was therefore sampled from the catheterized auricular artery approximately half way through the procedure (arterial blood gases were analysed at the same time).”

From their results it is quite apparent that the V-gels were easier to place then intubating rabbits.

From our experience sometimes they do not produce a tight seal around the larynx (leak) and may make ventilation difficult.

Perhaps one of the suggestions for future procedures would be intubating rabbits for these kind of procedures. This is what would be done in clinics.

We agree with the reviewer. Would the reviewer suggest to add this information to the conclusions?